# The conservation of human functional variants and their effects across livestock species

Rongrong Zhao[1], Andrea Talenti [1], Lingzhao Fang[1], Shuli Liu[2], George Liu[3], Neil P. Chue Hong [4], Albert Tenesa [1], Musa Hassan[1] & James G. D. Prendergast [1✉]

Despite the clear potential of livestock models of human functional variants to provide important insights into the biological mechanisms driving human diseases and traits, their use to date has been limited. Generating such models via genome editing is costly and time consuming, and it is unclear which variants will have conserved effects across species. In this study we address these issues by studying naturally occurring livestock models of human functional variants. We show that orthologues of over 1.6 million human variants are already segregating in domesticated mammalian species, including several hundred previously directly linked to human traits and diseases. Models of variants linked to particular phenotypes, including metabolomic disorders and height, are preferentially shared across species, meaning studying the genetic basis of these phenotypes is particularly tractable in livestock. Using machine learning we demonstrate it is possible to identify human variants that are more likely to have an existing livestock orthologue, and, importantly, we show that the effects of functional variants are often conserved in livestock, acting on orthologous genes with the same direction of effect. Consequently, this work demonstrates the substantial potential of naturally occurring livestock carriers of orthologues of human functional variants to disentangle their functional impacts.

[1] The Roslin Institute, University of Edinburgh, Easter Bush Campus, Midlothian EH25 9RG, UK. [2] Westlake Laboratory of Life Sciences and Biomedicine, Hangzhou, Zhejiang 310024, China. [3] Animal Genomics and Improvement Laboratory, Henry A. Wallace Beltsville Agricultural Research Center, Agricultural Research Service, Agricultural Research Service, USDA, Beltsville, Maryland 20705, USA. [4] EPCC, Bayes Centre, 47 Potterrow, Edinburgh EH8 9BT, UK. ✉email: james.prendergast@roslin.ed.ac.uk

Animal models are widely used across the biological sciences. From the development of vaccines and use as models of human diseases, to addressing fundamental questions about human biology. Importantly animal models provide the ability to test the effect of manipulating key variables in a controlled fashion, in ways that are not possible in human populations. For example, by altering the genome via genome editing. The introduction of variants thought to be functional in humans into animal models enables a range of studies, from the characterization of their downstream impacts on the expression of genes, to how different alleles respond to different interventions such as drug treatments.

By far the most widely used mammalian animal models are rodents, due to their ease of handling and short generation times. But rodent models have several limitations. Most importantly humans and rodents are physiologically very different, with the pathogenesis of diseases often differing substantially between the species. This has been proposed as a key driver of why less than 8% of cancer studies that are based on animal models result in a clinical trial[1]. Furthermore, the sizes of rodent organs poorly match those of humans, and it is difficult to serially sample rodent models due to their smaller size. Although the use of primate models can overcome many of these limitations their use is limited by both cost and ethical considerations[1]. For these reasons livestock species have been proposed as more effective animal models in many scenarios[2,3]. Pigs in particular have a similar size, physiology and anatomy to humans[4], and have been shown to have more similar gene expression patterns to humans than rodents[5]. As a result they are increasingly used in translational research, from toxicology testing of pharmaceuticals to the development of transgenic models of human diseases ranging from cystic fibrosis and diabetes to neurodegenerative disorders[6]. However, livestock models of human functional genetic variants have major drawbacks: they are expensive and time-consuming to generate. As well as the substantial time and costs associated with generating and implanting the genome edited embryos, it is necessary to maintain the mothers through long pregnancies in areas suitable for genetically modified animals, with no prospects of recouping the costs through selling the animals afterwards. There are also further ethical considerations to such transgenic projects, with the public often skeptical of the merits of artificially introducing human variants into other species.

Therefore, despite the clear merits of being able to assay the effects of human functional variants in livestock models, transgenic experiments come with several obstacles. Even among mice, the number of truly humanized models, i.e., where the directly orthologous mouse base or sequence has been altered to match that in humans, is low. Traditionally transgenic mouse models involve the random insertion of transgenes into the genome, meaning they lose their wider genomic context and potential impacts on downstream functions and mechanisms[7]. To properly model human functional variants, the same changes need to be made at orthologous locations, with both alleles present among the animal model.

A relatively under-explored alternative to the *de novo* generation of animal models is the study of natural orthologues of human functional variants. The 1000 bulls project alone identified over 84 million cattle single nucleotide polymorphisms[8], meaning approximately 1 in every 32 bases in the cattle genome is polymorphic. This though is potentially an underestimate of the expected probability of a human variant having a cattle orthologue, as polymorphisms are known to be dependent on the underlying sequence. For example, CpG sites are known to be susceptible to deamination, likely raising the probability of such sites being polymorphic across species. This suggests there are potentially many natural orthologues of human functional variants, meaning the effect of these variants can be studied in large mammalian models, potentially at scale, without resorting to transgenic approaches. Supporting this idea, although rare in the literature, some examples of functional variants being found naturally across different mammalian species have already been reported. For example, a missense change linked to coat colour found segregating among both dogs and water buffalo[9]. In recent work, non-naturally occurring coding changes in mice and zebrafish were compared to these found in humans, with orthologues of human pathogenic Clinvar variants shown to more likely also to lead to a detectable phenotypic change in zebrafish than other variants[10]. To date there has though been little genome-wide study of the natural orthologues of human functional variants and the conservation of their effects across mammals. In part this has resulted from the fact that the precise functional variant underlying most human quantitative trait loci and genome-wide association loci have been unknown. However, high resolution functional datasets and fine-mapping approaches have begun to disentangle causative variants from those simply in linkage disequilibrium[11,12].

Studying the impact of these functional variants has the potential to inform our understanding of phenotypes beyond just humans. This is because livestock species are not only good models for humans but the reverse is also true. Substantially more biological data and insights have been generated for humans than livestock, and characterizing how human functional variants effect corresponding phenotypes in livestock may provide insights into how to improve the production and health of domesticated animals. For example, the genetic basis of stature in cattle has already been shown to have parallels of that in humans[13,14], and better understanding functional variants linked to height could provide potential avenues for adjusting livestock body size.

The aim of this study was, therefore, to characterize the extent to which natural orthologues of human variants are found in domesticated species. Using machine learning we characterize the features associated with the presence of orthologues across species, investigate the presence of functional variants linked to diseases and traits across mammals, and determine where their effects on downstream phenotypes are conserved. We highlight how orthologues of human functional variants are likely a valuable resource to better understand the genetic basis of both human and livestock phenotypes.

## Results

**Extensive sharing of variants across species.** To investigate how often the same variants are found across species, we compared the 78 million human SNPs identified in the 1000 genomes cohort of ~2500 diverse individuals[15] to the variants identified in cohorts of 477 cattle[9] and 409 pigs[16]. In total 35 and 34 million of the human variants could be mapped to an orthologous location in the pig and cattle genome, respectively (Fig. 1a). Of these 3.7 and 3.0% overlapped an orthologous variant segregating in one of these other species, with 55.5 and 55.8% of these showing the exact same allele change. Consequently over 1.1 million human variants have a direct orthologue in at least one of these two livestock cohorts. Intersecting the same human polymorphisms with variants in cohorts of two further domesticated species, 722 dogs[17] and 81 water buffalo[9], revealed that 1,651,728 are found in at least one of these four mammalian cohorts (Fig. 1b).

The number of variants shared across cohorts from different species is expected to be a function of the number of samples in each cohort. To characterize this relationship we randomly downsampled the pig and cattle cohorts and recalculated the observed overlap with the total set of human variants. As shown in Fig. 1a

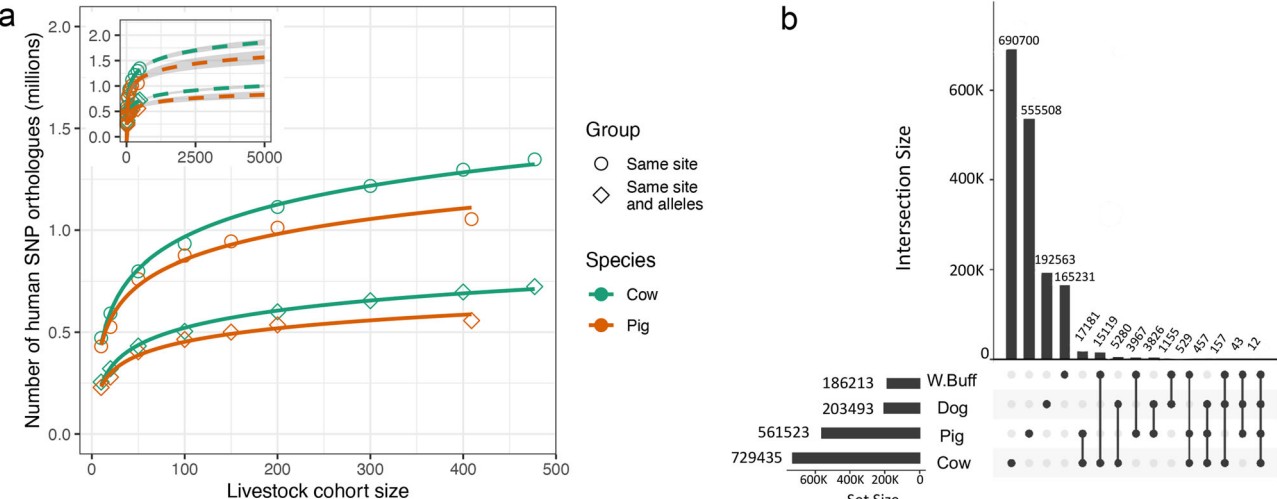

**Fig. 1 Frequency of variant sharing across species. a** Number of human (1000 genomes) SNPs that have a SNP at the orthologous location in each other species. Counts are broken down into where the SNPs have the same alleles across species (same site and alleles) or simply coincide, i.e. irrespective of allele change. The inset shows the number of orthologous SNPs expected in larger cohorts when extrapolating the curves. **b** The number of human variants overlapping a variant found in one or more other species with a matching allele change. The vertical bars indicate the number of human orthologues found across the species indicated by the dots below. The horizontal bars indicate the total number of orthologues in that species. The data underlying this plot can be found in Supplementary Data 1.

**Table 1 Variant annotations used in the modelling and their encoding method.**

| | Annotation | Data source | Encoding method | Number of features | Number of columns after encoding |
|---|---|---|---|---|---|
| Sequence conservation | phastCons100way | UCSC genome annotation database | - | 1 | 1 |
| | phastCons30way | | - | 1 | 1 |
| | phyloP100way | | - | 1 | 1 |
| | phyloP30way | | - | 1 | 1 |
| Variant position properties | Distance to CpG island | UCSC genome annotation database | - | 1 | 1 |
| | Distance to chromatin data[a] | Ensembl | - | 1554 | 1554 |
| | Distance to TSS[b] | Ensembl | - | 14 | 14 |
| | Distance to regulatory features[c] | Ensembl | - | 4 | 4 |
| | Chromosome | - | One-hot encoding | 1 | 22 |
| | Variant position | - | - | 1 | 1 |
| | Gene density (per megabase) | Ensembl | - | 1 | 1 |
| VEP annotations | Consequence | Ensembl | One-hot encoding | 1 | 34 |
| | Allele frequency[d] | | - | 6 | 6 |
| Sequence context | Allele change | Ensembl | Self-defined encoding | 1 | 8 |
| | 5-mer flanking sequence | UCSC genome annotation database | | 1 | 20 |

[a]Distance to 1554 different chromatin data types from Ensembl (regulatory build of hg38).
[b]Distance to TSSs within 14 common biotypes (e.g. protein coding, lncRNA etc. Those with a frequency in the genome of >= 1000).
[c]Regulatory features include enhancer, promoter, CTCF binding site, TF binding site.
[d]A total of six allele frequencies from the 1000 genomes combined population and the African, American, East Asian, European and South Asian populations separately.
In total 1589 features were used in the machine learning models to assess whether they were linked to the probability of a variant being found across species. These broadly fell into one of four categories and after encoding (conversion of categorical data to integer format) there were a total of 1669 features tested in the modelling.

the number of variants overlapping the human dataset had not plateaued for either species, suggesting larger cohorts would continue to identify even more orthologues of human variants. For example, extrapolating the results to 5000 samples in corresponding cohorts suggests over 840,000 pig and 1,000,000 cattle orthologues of human variants would potentially be detected (Fig. 1a). As expected, sample diversity/relatedness is also an important factor with more diverse cohorts leading to more orthologous variants being identified (Supplementary Fig. 1). This suggests exact orthologues of several million human variants are naturally segregating among livestock species.

**Modelling the distribution of shared variants across the genome.** Using 1589 different annotations (Table 1), including sequence conservation, chromatin context, and the distance to genome features such as genes, we investigated whether these features were linked to whether human variants had livestock orthologues. Several factors were associated with the probability of a human SNP having a cattle orthologue, including their distance to known genes and chromatin, and sequence context. For example, human variants with a cattle orthologue are more likely to involve a C to T change than those without a corresponding cattle orthologue (Fig. 2a). C to T changes in mammalian

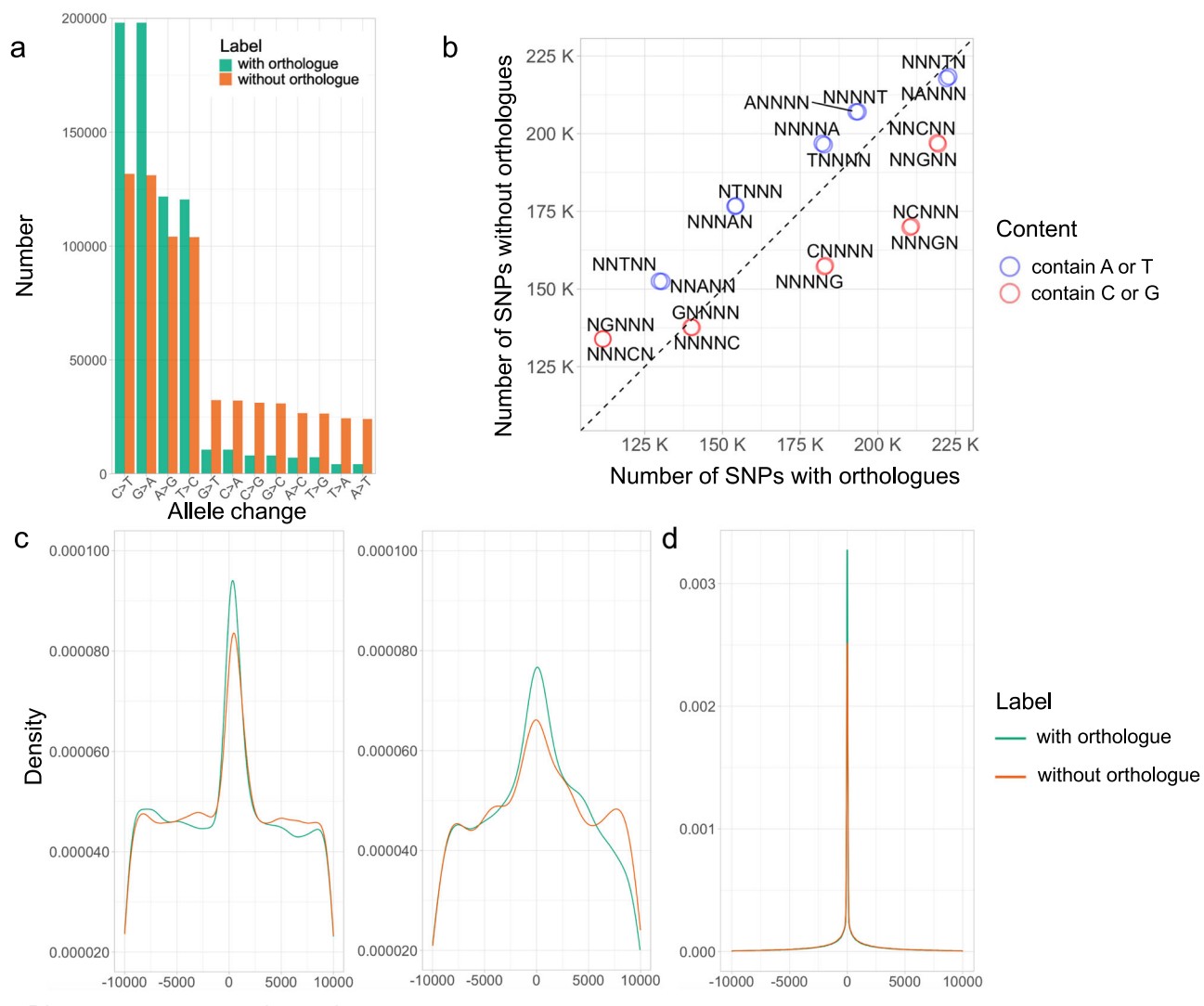

**Fig. 2 The characteristics of human variants with cattle orthologues.** The association of 1,395,750 human variants with different genomic features is shown (697,875 human variants with orthologues in cattle and an equally sized random sample of 697,875 human variants without known orthologues in cattle). **a** Number of variants with or without orthologues by their observed allele changes (reference > alternative). The data underlying this plot can be found in Supplementary Data 2. **b** Number of SNPs with different 5-mer flanking sequences among variants with or without orthologues. Each circle represents a 5-mer flanking sequence with a specific base at a certain position, and the circle color indicates whether the specific base is C/G or A/T. The black dashed line represents parity, i.e., where the number of SNPs with orthologues equals the number of SNPs without known orthologues. All the 5-mer sequences are significantly different between the groups at a $P$-value less than $2.2 \times 10^{-16}$ (Chi-Squared test). **c** Density plots of distances of variants with or without orthologues to processed pseudogenes and snoRNAs (plot restricted to within 10 kb). Distances of variants to processed pseudogenes and snoRNAs are different between groups at $P$-values less than $3.2 \times 10^{-5}$ (Two-sample Kolmogorov-Smirnov test). **d** Density plot of distance between variants with or without orthologues to chromatin regions marked by H3K9ac in the human H9 cell line (plot restricted to within 10 kb). Distance to these regions is different between groups at $P$-value less than $1.8 \times 10^{-3}$ (Two-sample Kolmogorov-Smirnov test). The data underlying this plot can be downloaded from https://doi.org/10.6084/m9.figshare.20401851.

genomes are commonly caused by the known hypermutability of CpG sites, whereby CpG sites are highly susceptible to deaminate to TpG[18]. The elevated mutation rates of these sites consequently likely increases the chance of the same change occurring across lineages. More generally, human changes with a G:C base pair within their 5-mer flanking sequence are more likely to have a cattle orthologue than those with an A:T base pair at the same position (Fig. 2b). A notable exception to this is where a guanine is found 5 prime of the human SNP site, with such changes less likely to have an orthologous SNP at the same position in cattle (Fig. 2b). Variants with orthologues are also more likely to be

enriched near specific genes such as processed pseudogenes and snoRNA, and around certain chromatin marks (Fig. 2c, d).

We investigated the extent to which it is possible to use these genomic annotations together to predict whether a human variant will have an orthologue in a livestock species. To do this, we used 140,000 human variants with or without a cattle orthologue and trained three tree-based machine learning models (Random forest, XGBoost and CatBoost, see methods) on the 1589-human genomic features (Table 1). To compare the performance of these models at discriminating human variants with and without cattle orthologues we tested the models on a further 60,000 human

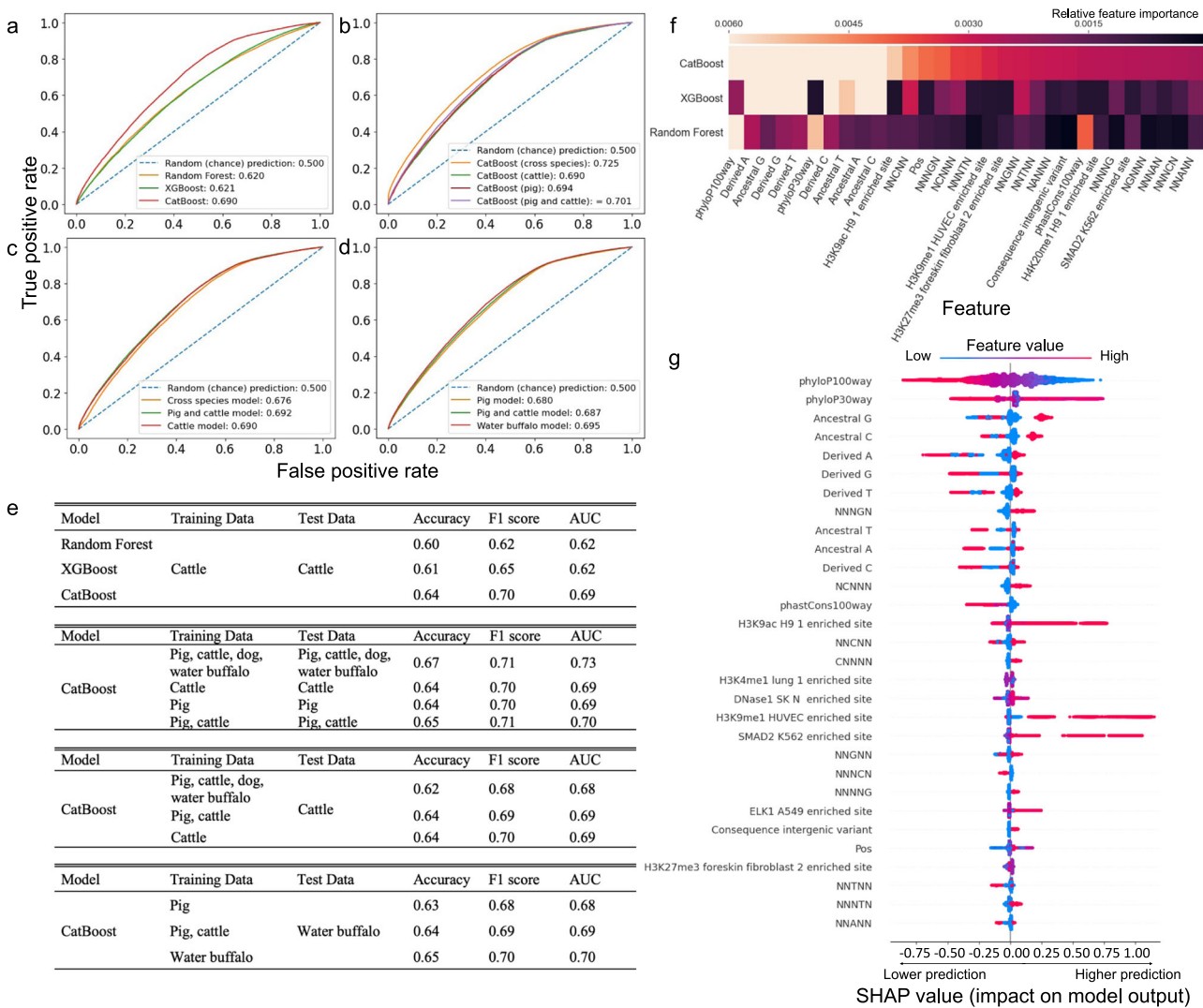

**Fig. 3 Machine learning models of orthologous variants. a** Receiver operating characteristic (ROC) curves of Random Forest, XGBoost and CatBoost models trained and tested using human variants with and without orthologues in cattle. The numbers in the legend are area under the receiver operating characteristics (AUC) scores of the different models. AUC reflects a model's general ability of distinguishing between the classes, the closer to 1 the better the model performance. Values of 0.5 would reflect a model not able to differentiate between variants with and without orthologues. **b** ROC curves of CatBoost models trained and tested using human variants with and without orthologues in cattle; pig; pig or cattle; pig, cattle, dog or water buffalo (cross species). **c** ROC curves of CatBoost models trained using human variants with and without orthologues in cattle; pig or cattle; pig, cattle, dog or water buffalo, but tested using human variants with and without orthologues in cattle. **d** ROC curves of CatBoost models trained using human variants with and without orthologues in pig; water buffalo; pig or cattle, and tested using human variants with and without orthologues in water buffalo. **e** Summary statistics of the experiments. Tables correspond to the same analyses shown in plots A to D respectively in the same order. **f** Feature heatmap of CatBoost, XGBoost, Random Forest models trained and tested using human variants with and without orthologues in cattle. Thirty important features are included in the figure, with lighter color indicating greater importance in that model. **g** SHAP summary plot[52] of the CatBoost model trained using human variants with and without orthologues found in any of cattle, pig, dog and water buffalo. A SHAP plot shows the relationship between feature values and their impact on model predictions. Features are ranked in descending order according to their importance on the left. The color represents low (in blue) and high (in red) value of the feature and the effect of their values on the output of the model is reflected by their positions on the x-axis. For example, lower phyloP100Way conservation scores (as indicated in blue) are associated with an increased probability of a human variant having an orthologue in another species (being higher on the x axis). The SHAP values underlying this plot can be downloaded from. The data underlying Fig. 3 can be downloaded from https://doi.org/10.6084/m9.figshare.20401851 and https://doi.org/10.6084/m9.figshare.20730370.

variants that had not been included in the original training data. As shown in Fig. 3a, e, CatBoost outperformed the other models with an area under the receiver operating characteristics (AUC) score of 0.69, an accuracy of 0.64 and an F1 score of 0.70. These results suggest that the genomic annotations of the variants contain discriminating information that makes it possible to identify which human variants have a higher probability of having an orthologue in another species.

Models trained on human variants with orthologues in other species, such as pig, could predict the presence of orthologues in these other species with similar accuracies as the cattle specific models (Fig. 3b, c, e). Likewise models trained on human variants with orthologues in given species were largely as accurate at predicting orthologues in completely different species (Fig. 3d, e). This suggests the features associated with orthologous variants are fundamental across mammals.

Comparison of the top 30 most important features of the three different modelling approaches shown in Fig. 3a found that the allele change, 5-mer flanking sequence and conservation score (phyloP100way) were consistently three important features (Fig. 3f). Figure 3g shows the top 30 most important features of the cross-species model, i.e. trained using human variants with an orthologue in any of the tested livestock species, and how their values affect the predictions of the model. Sequence conservation is the most important variable, with human variants in less-conserved regions more likely to have an orthologue in another species. This is consistent with mutations in these regions less often being removed, increasing the probability of the same change occurring in different mammalian lineages. As well as the type of base change the flanking sequence disproportionately contributes to the model performance, with a G base at the position immediately downstream of the polymorphic site (NNNGN) associated with an increased probability of the same change being observed in the other species, consistent with the preferential deamination of CpG sites.

**Animal models of human pathogenic variants**. Although over a million human variants have a livestock orthologue (Fig. 1), the modelling results above highlight that these disproportionately fall in less conserved, and consequently most often non-functional, genomic regions. This raises the question as to how many naturally occurring livestock models of functional human pathogenic variants exist. To characterize specifically how many human pathogenic mutations are segregating in other livestock species we first extracted 89,158 SNPs from the human Clinvar[19] database labelled as "pathogenic" or "likely pathogenic". Being mostly found in conserved coding regions, the overwhelming majority (99.4% and 94.1%) of these variants could be successfully mapped to an orthologous position in the Cow (BosTau9) and Pig (SusScr3) genomes. Using the data from the same cow and pig cohorts we identified how often these variants overlapped an orthologous variant segregating in one of these other species. In total 1290 Clinvar variants overlapped a variant in the cow dataset and 767 in the pig cohort, of which 253 and 212 respectively also showed the exact same allele change observed in humans. In agreement with the modelling results, these numbers differ from those expected from the background numbers. Not only is the number of Clinvar variants with an orthologue in one of these livestock species substantially lower than expected given the number of all human variants with an orthologue in pig or cattle, but also where a variant does segregate at the orthologous position it is less likely to show the same allele change. In total Clinvar variants are approximately three times less likely to have a variant at the orthologous position in either pig or cattle than expected from the frequencies in the 1000 genomes cohort, and approximately seven times less likely of having one displaying the same allele change (Supplementary Fig. 2). These results are consistent with these changes being deleterious as indicated, and selection preferentially removing them across species.

Orthologous variants, even with the same allele change, may not have the same impact on genes, if for example the gene structure and codons have changed between species. As shown in Fig. 4a, 80% of the 103 cattle orthologues of human Clinvar variants leading to a missense change show the same missense change across both species. A further 13% are missense in both species but involving different amino acid changes. Only 3.9% of the human missense variants are predicted to be synonymous in cattle, suggesting the consequence of human missense changes is most often conserved across these mammals, with similar patterns observed in pigs (Fig. 4b). However, of 115 human Clinvar variants predicted to lead to the introduction of a stop

codon, only 22% also lead to a stop gained change in cattle, with the majority (63%) predicted to just lead to an amino acid change due to a difference in the codon between species. This may represent a true difference in the impact of these variants between species but may also sometimes reflect the comparatively poor annotation of gene isoforms in livestock species. Of note, 33 cattle and 20 pig variants lead to the same protein impact as their orthologue in humans despite involving a different allele change (Fig. 4c, d). Consequently, although rare, variants do not necessarily need to show the same allele change to have a conserved impact.

These data highlight that there are existing animal models available for at least several hundred human Clinvar variants, including those linked to a variety of important phenotypes such as cancers and Parkinson's disease (Supplementary Data 4). Interestingly, Clinvar variants linked to certain traits are more likely to be found across species. This includes those linked to biotinidase deficiency (Chi-squared test $P < 1 \times 10^{-7}$), neurofibromatosis ($P = 1.8 \times 10^{-5}$) and glycogen storage in cattle ($P = 3 \times 10^{-7}$) and factor VII deficiency in pigs ($P < 1 \times 10^{-7}$). Of 23 known human biotinidase deficiency variants, four (17%) have a direct orthologue showing the same allele change in cattle. This is despite only 1.5% of all lifted Clinvar variants having a cattle orthologue. All four of these variants are missense SNPs showing the same amino acid change in both species, with one of the mutations having risen to a minor allele frequency of 22% in cattle despite being found at a frequency of only 0.002% in humans, meaning studying its impact may be easier in cattle populations. Supplemental biotin is often fed to cattle as it is thought to improve hoof health and increase milk production[20] and these variants are consequently also strong candidate functional variants for further investigations for improvement of these cattle traits.

**Animal models of common variants of polygenic diseases and traits**. We next investigated whether there are potential existing livestock models of common human variants linked to polygenic diseases and traits. To do this, we obtained 2240 fine-mapped SNPs linked to 47 different traits in the UK biobank cohort[21]. In total 58 of these variants had a direct orthologue segregating in either pigs or cattle. Interestingly variants linked to height in humans were significantly more likely to have a direct orthologue in cattle than other traits, with over a quarter of variants (11 out of 43) that were found in both species with the same alleles being linked to this phenotype (Fig. 5 and Supplementary Data 5). This is compared to only 13.6% (341 out of 2513) of the variants successfully mapped between the species being linked to this trait (two-tailed Fisher's exact test $P = 0.040$). Of these 11 variants, 3 are missense changes (rs154001, rs61735104, rs79485039), with each leading to the same amino acid change in both species. These amino acid changes fall in FGFR3, KIAA1614 and FBN2, with a further gene, *FOXM1*, having a variant (rs28990715) at orthologous positions in both species that leads to the same amino acid change despite having different allele changes. 10 of the 11 human variants with cattle orthologues were in the Gene Atlas UK Biobank[22] results and are together, under certain assumptions, associated with a predicted 2.7 cm variation in human height. This corresponds to around ~1/3 of a standard deviation of the human heights in the UK Biobank cohort. The FGFR3 change alone is associated with a ~1 cm difference in standing height between opposing homozygotes. Mutations in FGFR3 underlie 99% of cases of human achondroplasia that affects bone development and leads to short stature, but the role of this gene in cattle stature is less well characterized[23]. Potentially, in part, because all 11 variants are rare in cattle (10 with a

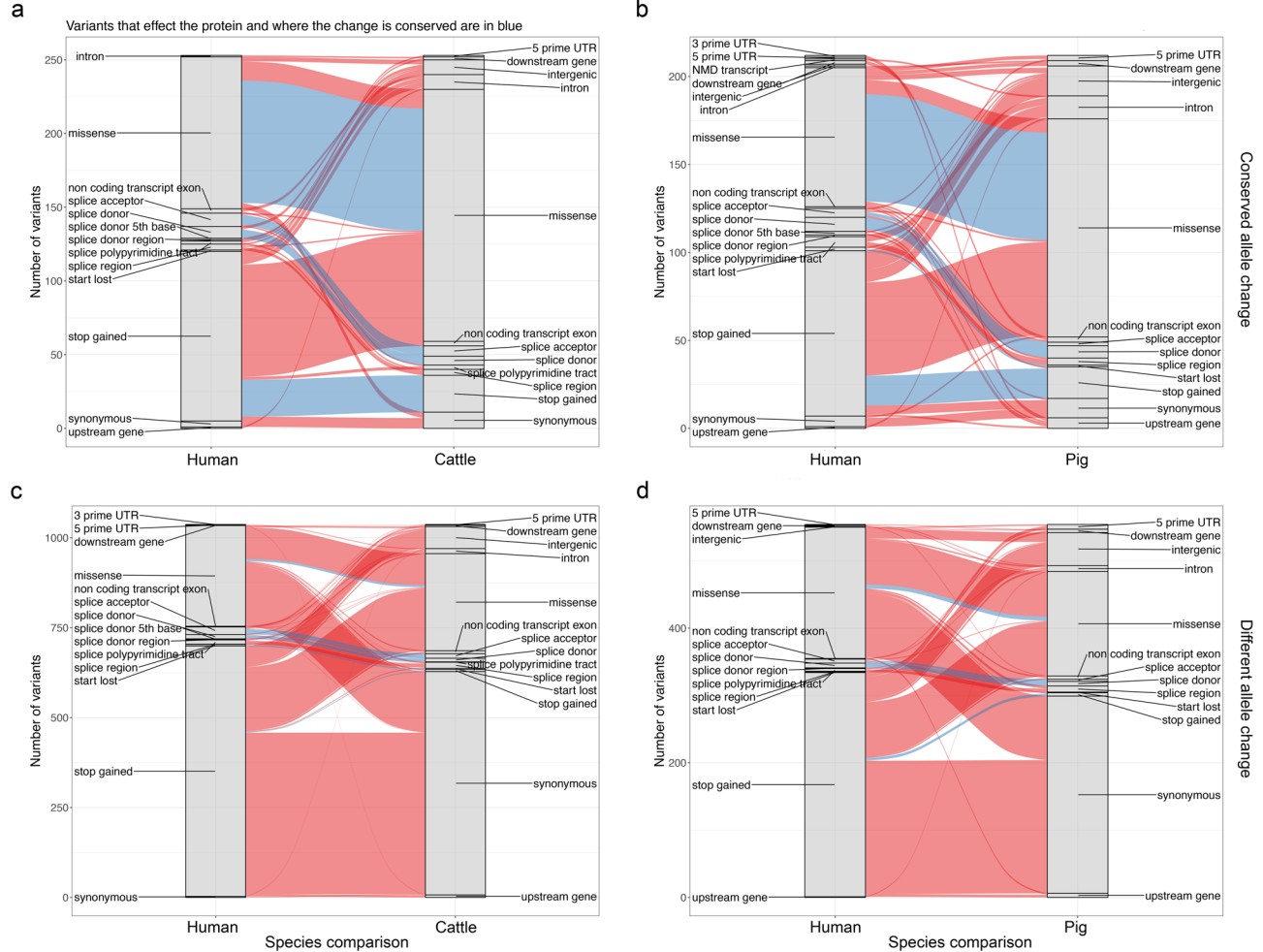

**Fig. 4 Conservation of impacts of orthologues of Clinvar variants.** Each plot shows the consequence of variants in humans (on the left) and the consequence of variants found at the same position in the pig or cattle genome on the right. The ribbons connect sets of variants, with the width of each ribbon indicating the number of variant pairs with the given combination of consequences across the two species. Ribbons in blue indicate where variants have potentially conserved impacts on the protein across species. **a** The conservation of impact of genic orthologous variants across human and cattle where variants show the same allele change. **b** The same as (**a**) but for human-pig orthologous variants. **c** Conservation of impact of variants across human and cattle where their locations are orthologous but they show different allele changes. **d** The same as (**c**) but for human-pig orthologous variants. The data underlying this figure can be found in Supplementary Data 3.

frequency of less than 1%, with 1 with a frequency of 6%) and would be unlikely to be detected in a standard cattle GWAS, but these variants are consequently strong candidate functional rare variants for contributing to variability in cattle height due to their strong associations with this trait in humans that could be exploited to alter cattle stature.

**Conservation of regulatory variation.** Most variants linked to important complex phenotypes are thought to be regulatory rather than coding[24]. To investigate whether regulatory variants are conserved across species we obtained the location of fine-mapped regulatory SNPs from the human GTEx[25] dataset. These human regulatory variants had been fine mapped using three different approaches; CAVIAR[26], CaVEMaN[27] and DAP-G[28], and we took the superset of SNPs across all three. We then extracted the associations with orthologous genes of variants found at the orthologous location in cattle from the cattleGTEx project, who defined eQTLs across 23 different cattle tissues and cell types[29]. In total 221 of the human-fine mapped variants had a matching cattle variant in the cattleGTEx data that had been tested against the same orthologous gene in at least

one tissue (Supplementary Data 6). Ignoring the allele change of the variants this number increases to 469. As shown in Fig. 6a, these cattle variants at the orthologous position of the human fine-mapped variants are more likely to show an association (i.e. have a smaller *P*-value) than randomly sampled gene-variant pairs from the cattleGTEx cohort. This suggests these variants are often regulatory across both species. Notably this was largely observed whether restricting the cattle variants to those showing the same allele change as the human variant or not, with only a slight enrichment of smaller P values among the former group in some tissues (Fig. 6a). This suggests simply disrupting the same regulatory site may often be sufficient to affect the gene's expression across species in many cases.

The direction of effect of conserved regulatory variants were more likely to be conserved across the two species (Fig. 6b), i.e., the same alleles are associated with increased or decreased expression across species. This confirms that the effect of predicted functional variants appear often conserved. This is despite the different linkage disequilibrium patterns between the species, that may be expected to disrupt any conservation of direction of effects if these variants were not functional.

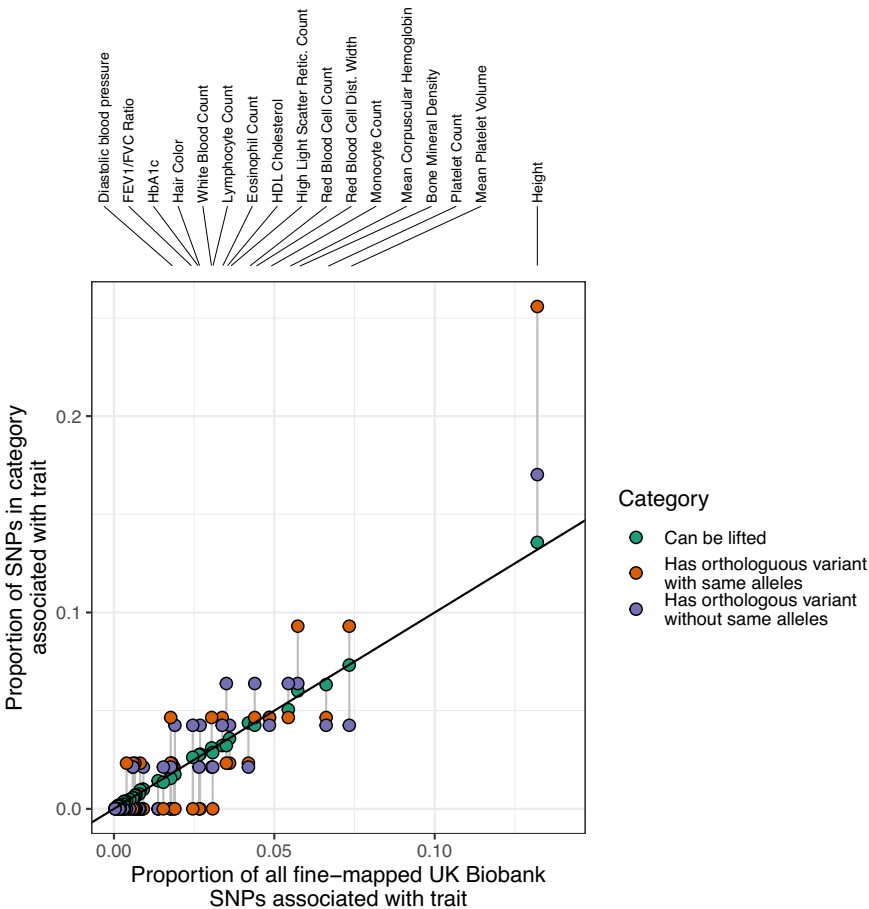

**Fig. 5 Orthologues of fine-mapped variants linked to traits in the UK Biobank.** All 2240 fine-mapped UK biobank variants were lifted over to the cattle genome and the number that overlapped a variant in the cattle genome with and without the same alleles were determined. The Y-axis shows the proportion of all SNPs that could be lifted that are associated with each trait (green), that have an orthologue with the same alleles (orange) or that have an orthologue with different alleles (blue). These values are compared to the expected values (X-axis) as represented by the proportion of all fine-mapped variants linked to the given trait. Circles corresponding to the same trait are connected by vertical grey segments, with the trait indicated above for those traits with at least 55 fine-mapped variants. The proportion of all lifted variants that were associated with a given trait is strongly correlated with the original proportion of fine-mapped variants linked to that trait (green circles). However, variants with an orthologous cow variant are disproportionately associated with height, and in particular those with matching alleles in both species (orange circles). The data underlying this figure can be found in Supplementary Data 5.

Figure 7 shows examples of the colocalization of eQTLs across humans and cattle. rs115287948 is a missense variant in the *SIRPA* gene that was finemapped in the GTEx cohort as a causative regulatory variant (probability > 0.5) linked to the expression of *SIRPB1* across a range of tissues including cultured fibroblasts and muscle (Fig. 7a–c). A direct orthologue of this variant is also found at a co-localised eQTL in cattle muscle (Fig. 7d) displaying an association with the same gene with the same direction of effect.

rs2230126 is a variant falling within an alternative promoter of *TAF1C* with which it is a fine-mapped human regulatory variant (probability > 0.95) in a range of tissues including subcutaneous adipose (Fig. 7e, f). An orthologue of this variant is also found in cattle and is the lead eVariant for the same gene in Nellore muscle tissue (Fig. 7g).

We explored why some variants conserved in both species may not show evidence of impacting gene expression in cattle. Figure 8 illustrates predictions from the Enformer human deep learning model[30], that predicts transcriptional potential and chromatin states from DNA sequence alone. As shown in Fig. 8a, b, Enformer predicts that the alternate allele of the rs10849334 variant is associated with reduced expression, specifically of an alternate, shorter isoform of the *NINJ2* gene. The predicted transcriptional

potential of the TSS of longer isoforms of the *NINJ2* gene are unaffected by this variant. When the orthologous cattle DNA segment is run through Enformer the predicted transcriptional potential of the TSS at these longer isoforms remains, but the CAGE peak at the promoter of the shorter isoform is completely abrogated. Consistent with this we could also not find any evidence of a cattle orthologue of this shorter isoform in the public databases. Consequently, the lack of evidence of the cattle orthologue of rs10849334 affecting the expression of *NINJ2* is potentially due to the human variant specifically regulating this shorter isoform, that is absent in cattle due to sequence divergence in this locus.

In summary, although the effect of some variants do not lift over across species, potentially due to, for example, changes in the usage of isoforms between species, a range of human regulatory variants have orthologues in cattle that often have conserved effects and can consequently be used to provide insights into their mechanisms of gene regulation.

## Discussion
In this study we have demonstrated how millions of orthologues of human variants exist in domesticated species, including

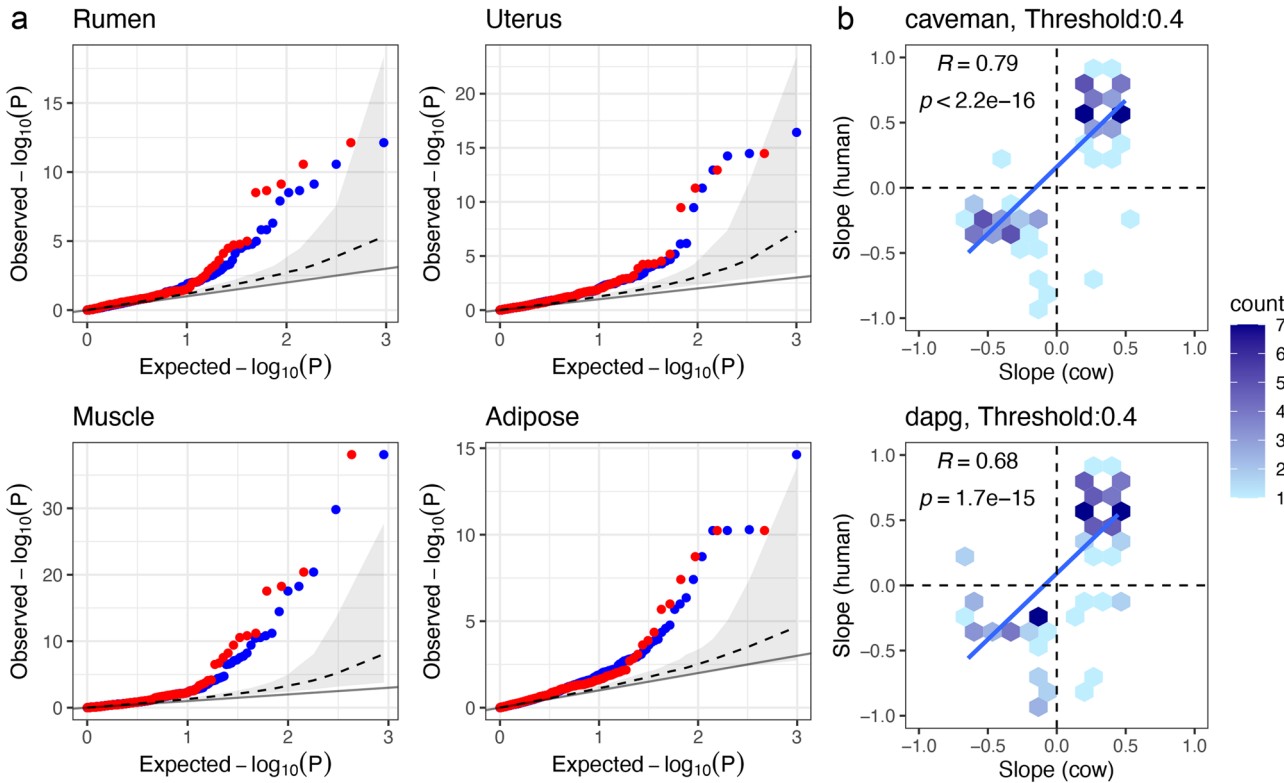

**Fig. 6 Conserved effects of regulatory variants across humans and livestock. a** Quantile-quantile (Q-Q) plots of observed and expected cis-eQTL *P* values of cattle variants that are direct orthologues of human fine-mapped regulatory variants. The blue points represent the observed and expected *P* values of the cattle variant's association with the expression level of the cattle orthologue of the corresponding human gene in four cattle tissues irrespective of its allele change. The red points are the same after restricting to these variants exhibiting the same allele change as observed at the human SNP. The grey dashed line and grey ribbon represents the median and 95% confidence interval obtained when randomly sampling the same number of variants as shown by the blue points from all cattle variants tested (irrespective if have a known human orthologous variant or not) 1000 times. The line of parity (solid black) is also shown. This illustrates cattle orthologues of human fine-mapped regulatory variants are more likely to show evidence of also being linked to the orthologous gene's expression across different tissues. **b** Comparison of the slopes (direction of effects) of eVariants across species. The slopes of human fine-mapped regulatory variants (using caveman or dap-g approach) were compared to the slopes observed for their orthologues if they also had a significant cattle GTEx association with the expression level of the orthologous gene in cattle. The slope represents the impact on the gene's expression of increasing the dosage of the same allele in both species. Note the same eVariant can be found in multiple tissues and can therefore be represented multiple times in this plot. In total there are 83 and 106 human-cattle association pairs in the caveman and dapg plots, involving 43 and 57 distinct human eVariant-gene-tissue associations. The significant positive correlation remains if only one entry for each human eVariant-gene-tissue association is retained. This agreement in direction is seen despite not restricting to comparing the effects to the same tissues across species, i.e., the direction of effect is generally conserved across tissues as well as species. The data underlying this figure can be found in Supplementary Data 6.

hundreds of orthologues of fine-mapped functional variants linked to diseases and phenotypes. These are consequently readily accessible large animal models of important human variants, that can potentially be studied at scale without the time and costs associated with transgenic approaches. Importantly we show that orthologous regulatory variants most often have conserved directions of effect across humans and cattle, suggesting their downstream effects can be effectively studied in these species.

Variants shared across humans and domesticated species are not restricted to one type of trait, but are linked to a wide spectrum of phenotypes. From rare, monogenic disorders such as cystic fibrosis to highly polygenic traits such as height. However, variants associated with particular phenotypes are found across species more often than expected. For example, the 4 out of 23 variants associated with biotinidase deficiency that have a direct orthologue in cattle. It is unlikely the co-occurrence of these variants is purely due to, for example, a higher mutation rate around the gene linked to this phenotype, as none of the other three species studied carry even one orthologue of these variants. This suggests there is a preferential overlap of variants linked to

specific phenotypes, including more polygenic phenotypes such as height. Consequently, although the sharing of the overwhelming majority of variants across species is likely the result of neutral processes, the disproportionate sharing of variants linked to certain phenotypes potentially reflects selection to preferentially maintain such variants arising in each species. This not only provides insights into the evolution of these species, but also potential candidate variants for livestock breeding programs. The increased number of human height associated variants with an orthologue in cattle likely reflects the selection for body size in domesticated animals. However, as these variants remain polymorphic, the selective sweep is incomplete, and they remain suitable targets for breeding programs. Although there would be potential ethical concerns of introducing human variants into livestock species to improve their production, the same is not true if the variant already exists in the species, as even editing the variant into another breed, would no longer come with the restrictions imposed on transgenic projects. Consequently, exploiting the large amount of data and studies on functional variants in humans, could potentially be leveraged to prioritise variants for testing their effects in livestock.

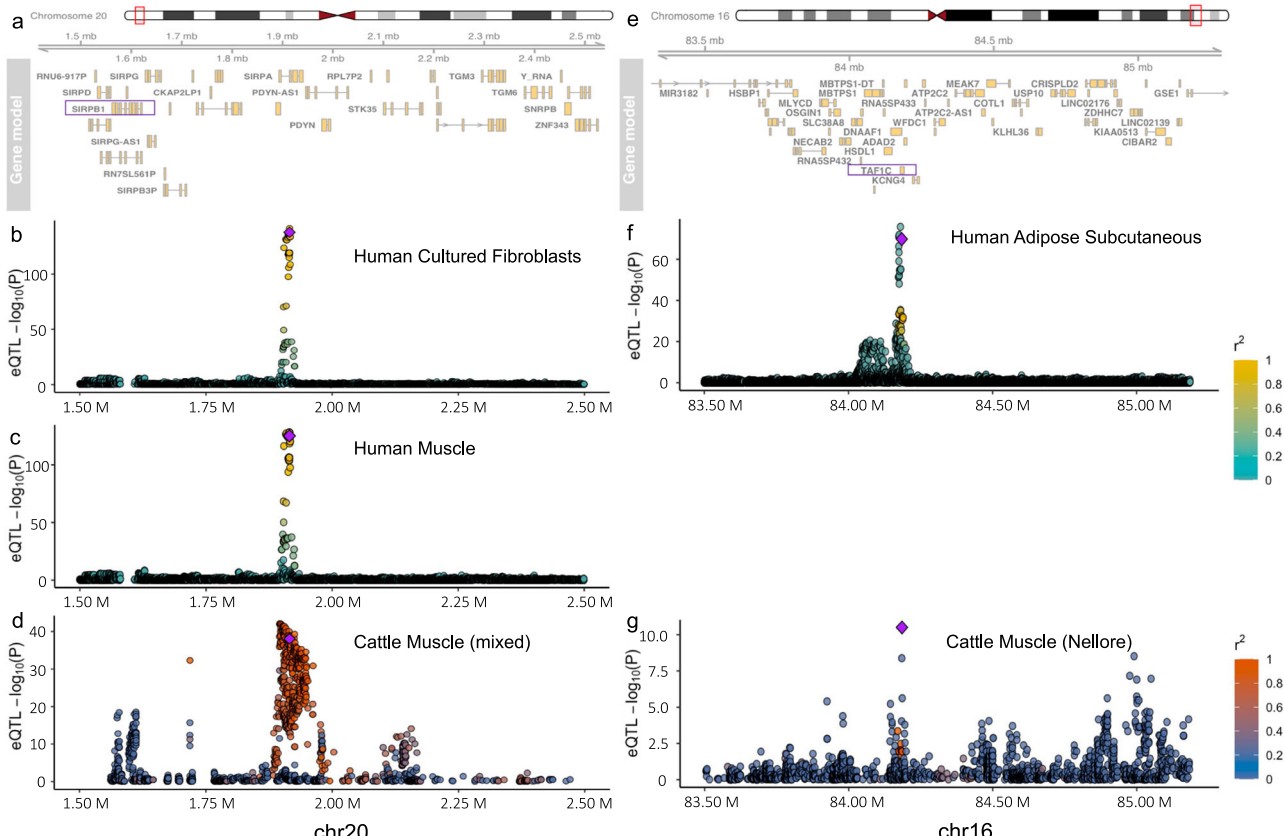

**Fig. 7 Colocalization of eQTLs across humans and cattle. a** The human gene neighborhood (hg38) of a shared eQTL, rs115287948, found across both humans and cattle. The gene regulated by the eQTL is indicated by a purple rectangle. **b** Strength of association of human variants with *SIRPB1* expression levels in cultured fibroblasts tissue. The fine-mapped regulatory variant, rs115287948, with a cattle orthologue is represented by the purple diamond. Other variants are colored according to their linkage disequilibrium ($r^2$) with this variant. **c** Strength of association of the same human variants but in muscle tissue. **d** Strength of association of variants with *SIRPB1* in cattle muscle tissue (mixed breeds). Each variant is plotted according to their orthologous position in the human genome and the variant with a fine-mapped orthologue in the human GTEx data is represented by the purple diamond. **e–g** same as **a**, **b**, **d** but for variant rs2230126 linked to the expression of *TAF1C* in different tissues. The data underlying this figure can be found in Supplementary Data 7.

Despite the hundreds of orthologues of functional variants identified, this number is likely a substantial under-estimate of the true number of shared functional variants. This is because for a variant to be tied to a phenotype it needs to not only be at a sufficiently high frequency in the population to be discovered, but also with suitable data and patterns of linkage disequilibrium to be fine-mapped. However, most functional variants are, by their nature one or more of: rare, non-coding or in regions of elevated LD, and are therefore difficult to tie to a trait. Looking at the effects of rare variants across species may though help increase the pool of individuals in which to study their potential role. Likewise looking across species has the advantage that allele frequencies and linkage disequilibrium patterns can differ substantially and may, therefore, help in fine-mapping approaches. Extending this further, such approaches may help validate fine-mapping methods, for example by characterising which fine-mapping approach better identifies variants whose impacts are subsequently shown to be conserved across species. This is illustrated by the comparison of fine-mapped regulatory variants, and their conserved direction of effect across species. Of the three fine-mapping approaches studied, CAVIAR fine-mapped variants showed the lowest conservation of effect direction. This may reflect where variants are not truly functional, with the different patterns of LD in the different species with the actual causative variant meaning their eQTL coefficients are less conserved.

A caveat to such cross-species comparisons of regulatory variants is that not only can it be difficult to directly match tissues between species, but that power also generally differs due to differences in sample sizes. Consequently, the fact that a variant doesn't also show evidence of being linked to a gene's expression in another species doesn't mean it is not a functional variant in both. Those variants we detect as being linked to gene regulation across species are likely those that are regulatory in multiple tissues, increasing the probability of us detecting it's association in at least one.

It is likely that few if any human polymorphisms with orthologues in these mammalian species arose prior to the divergence of the respective species, and to still be polymorphic down the independent lineages. Rather they will have largely arisen independently in each. This is supported by the fact that despite there being millions of orthologues of human variants segregating in other mammals, few are found in more than one other species. Only twelve sites were polymorphic across the five studied mammals. Shared variants are simply most often found at sites with the highest mutation rates and lowest levels of purifying selection, and therefore reflect the increased chance of these sites mutating and not being purged from the population in both species. This indicates that the normalised presence or absence of orthologous variants may provide an alternate metric of the selective pressure on genomic regions, as illustrated by the depletion of orthologues of Clinvar variants across species.

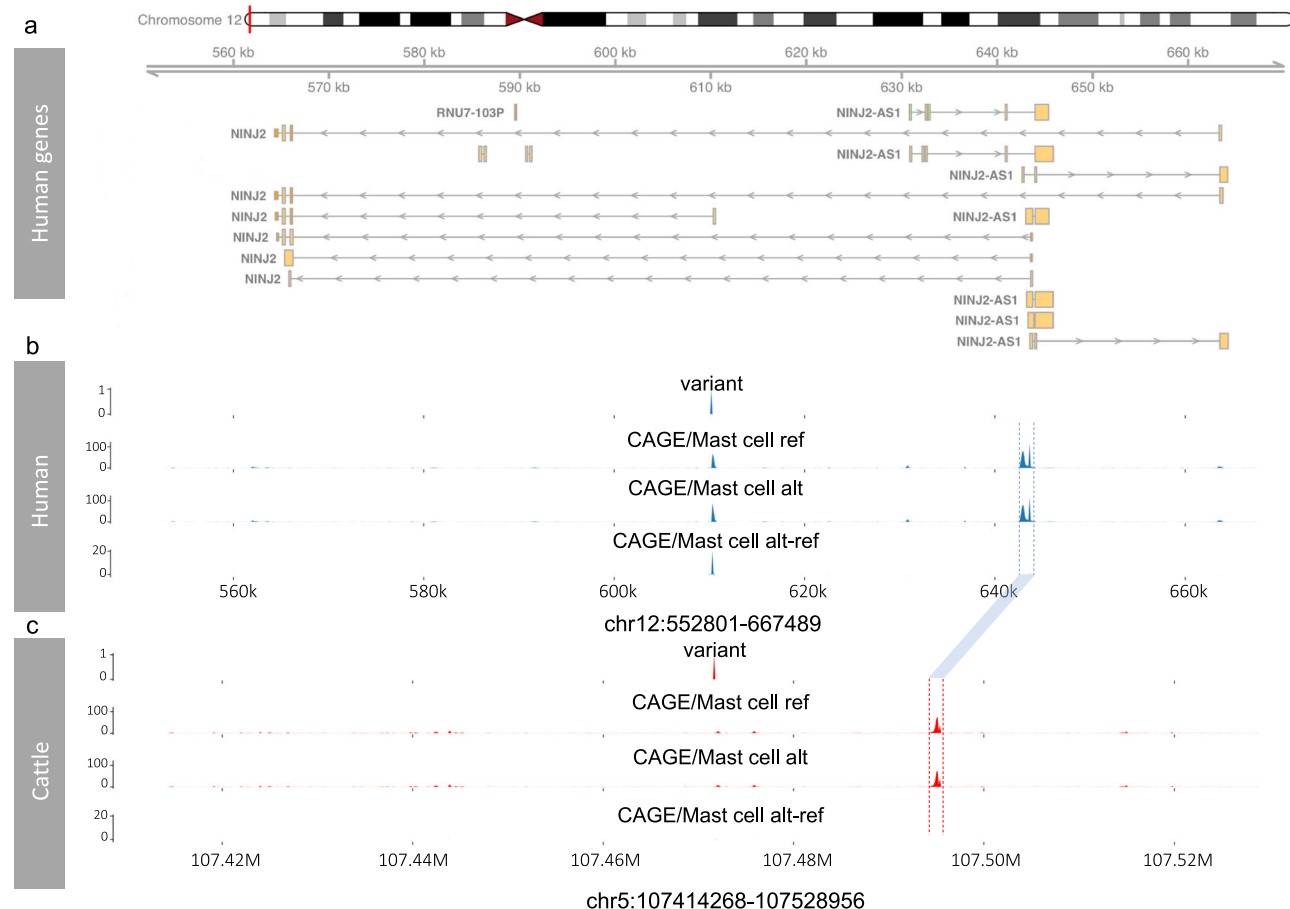

**Fig. 8 Enformer predictions for human and cattle. a** The human gene neighborhood of variant rs10849334. Different isoforms of the genes are included in the plot. **b** Enformer predicted Cap Analysis Gene Expression (CAGE) tracks of variant rs10849334 in mast cells (the track showing the largest human alt-ref difference). The first three tracks show i) the position of the target variant in the human genome, and the predicted CAGE levels for the ii) reference and iii) alternative alleles from Enformer, i.e., the predicted CAGE tracks obtained by taking the human DNA sequences containing reference and alternative alleles at the variant position as the inputs of the Enformer model. The final track shows the predicted difference between the CAGE levels from the reference and alternate sequences (alt-ref). As can be seen the only difference is observed specifically at the start position of the shorter *NINJ2* isoform in the centre of the plot. **c** Corresponding predicted CAGE tracks derived from the cattle sequences around the orthologous variant of rs10849334. The orthologous peaks at the TSSs of the longer *NINJ2* transcripts in cattle and human are indicated by the blue linking bar. However, no CAGE peak is predicted at the TSS of the shorter isoform of *NINJ2*, unlike with the human sequences. The data underlying this figure can be found in Supplementary Data 8.

Consequently the study of orthologues of human functional variants can be used across a range of studies. From understanding the biological mechanisms linking variants to important downstream phenotypes, to providing potential targets for livestock breeding and genome editing programs as well as understanding the selection pressures on our species.

## Methods

**Genotype datasets**. Previously published and filtered genotype data for five different species was used in this study. The genome-wide set of 78 million human SNPs from 2504 individuals was obtained from the 1000 genomes consortium[15]. The dog genotypes from 722 individuals were obtained from Plassais et al.[17]. The cattle and water buffalo genotypes of 477 and 79 individuals, respectively, were obtained from Dutta et al.[9], and the pig genotypes from across 409 individuals from the Genome Variation Map website[16]. All cohorts were subsequently restricted to biallelic SNPs only (cattle: 87,964,998; pig: 90,901,469; water buffalo: 37,682,631; dog: 73,906,017). For all sets of human variants, their positions were lifted to their orthologous positions in the pig (SusScr3), cattle (BosTau9) and dog (CanFam3) genomes using the UCSC liftover utility[31] with chain files available from the UCSC website. For the water buffalo[32], where no public chain file exists, we used the nf-LO pipeline[33] to perform the liftover. Sites that were lifted to more than one location were excluded. SNPs from other species were said to have the same allele change as human SNPs if found at the orthologous position with alleles that directly matched or that matched their complement. This therefore assumes the ancestral base in these conserved regions is the same across mammals.

To test the impact of relatedness on the number of orthologous variants found in cattle, we used the relatedness2[34] parameter in vcftools[35] to identify pairs of animals with a kinship coefficient greater than 0. Individuals in each pair were then iteratively removed till the kinship coefficient between all pairs of remaining animals was 0 or less.

**Clinvar and UK biobank analyses**. The location of variants potentially linked to human health were downloaded from Clinvar[36], which contains SNPs linked to different human clinical phenotypes. Restricting this set to those labelled as "pathogenic" or "likely pathogenic" left 89,158 SNPs with likely functional consequences. Potentially functional variants linked to polygenic traits were obtained from Weissbrod et al.[21]. This study produced a list of 3281 fine-mapped, potentially functional variants associated with 47 complex traits of which 2240 were SNPs. The locations of these sets of SNPs were intersected with those from other species to identify those segregating in other mammals as described above. To test whether Clinvar variants linked to particular phenotypes are more likely to segregate in another species than expected, we used a Chi-squared test to examine whether the proportion of successfully lifted Clinvar variants linked to a particular phenotype that overlapped a variant with matching alleles was significantly higher than that observed across all other phenotypes. To test whether UK biobank variants lifted to the cattle genome were disproportionately associated with height a Fisher's exact test was used, comparing the proportion of variants with an orthologous variant with the same alleles that were linked to human height versus

the proportion of successfully lifted variants linked to the same trait. To examine the impact of these orthologous variants on genes in humans, pigs and cattle, the variants were annotated using the Ensembl REST API in R, recording just the most severe reported consequence in each case.

**Regulatory variant analyses**. The GTEx v8 fine-mapped results for CaVEMaN, DAP-G and CAVIAR were downloaded from the GTEx portal. Together these reported 5,341,519 distinct tissue-gene-variant associations of which 2,145,167 could be lifted to an orthologous position in the cow genome. Upon filtering out variants that did not have a minimum probability > 0.2 in at least one of the datasets, this number reduced to 230,991 associations. These associations were then intersected with the cattleGTEx data to identify where an orthologous variant was significantly associated with the corresponding orthologous gene. Nominal P values were obtained as described in the original cattleGTEx paper[29] and the human-cow gene orthologues were obtained from Ensembl version 103[37].

To examine whether cattle orthologues of human fine-mapped eVariants were more likely to show evidence of also being significantly associated with the expression level of the same gene, we extracted their corresponding P values from the cattleGTEx data by tissue. The distribution of these P values were then compared to the distributions of *P*-values for the same tissue of the same number of variants sampled from the total cattleGTEx data 1000 times to produce the shaded confidence intervals in the Q-Q plots.

To conservatively estimate false discovery rates for the cattle eQTLs we used the same random samples. For each real eQTL P value we divided the average number of tissue-specific P values across the 1000 samples that had a P value as small or smaller by the corresponding number within the variants that were orthologues of human fine-mapped regulatory variants. This therefore corresponds to the approximate probability of having sampled a P value as small or smaller from the background list of all variants tested in the cattleGTEx project. This is conservative as a large number of the variants in this background list are eVariants. Therefore, this FDR corresponds to the false discovery rate above and beyond that expected given the number of regulatory variants in the background, and variants with a large FDR may still be regulatory variants.

To investigate why some regulatory variants shared across human and cattle may not have conserved impact on gene expression in cattle, we used Enformer, a deep learning architecture designed for predicting how DNA sequence influences gene expression[30]. We loaded the trained Enformer model, made predictions for reference and alternative alleles of each shared regulatory variant in both human and cattle, and obtained 5,313 predicted genomic tracks for each variant. The effect of each variant was evaluated by the difference between the reference and alternative predictions.

**Variant annotation and modelling**. The genome-wide set of 78 million human SNPs were annotated with 1589 features across four categories (Table 1), including sequence conservation, variants position properties, VEP[38] annotations and sequence context. For sequence conservation, we included 4 different conservation scores: phastCons100way, phastCons30way[39], phyloP100way and phyloP30way[40]. We downloaded bigWig files of these conservation scores from the UCSC genome annotation database[41] (hg38) and extracted the values at given positions using the pyBigWig python package[42]. To fully capture the position characteristics of the variants, we calculated the distance between the variants and different genome elements. We obtained the location of CpG islands from the UCSC genome annotation database[41], chromatin data (such as histone marks), TSS and regulatory features (enhancer, promoter, CTCF binding site and TF binding site) from Ensembl[37] (version 103). We used bedtools[43] closest command to calculate distances to CpG islands and chromatin data. The ChIPpeakAnno[44] R package was used for getting distances to the nearest TSS by biotype (only common biotypes were included, count > = 1000) and distances to various regulatory features. Then we used the VEP[38] command line tool to annotate the variants and get the allele frequencies and consequences of the variants. Instead of using Reference/Alternative alleles, we used Ancestral/Derived alleles for the allele change. The human ancestral genome (hg38) was downloaded from Ensembl[37] (version 103) and the bedtools[43] getfasta command was used to extract the ancestral base. To get the 5-mer flanking sequences centered on the target variants, we used the samtools[45] faidx command. Then we calculated gene density (per megabase) of each variant using the findOverlaps and queryHits functions in the GenomicRanges package[46].

Using these genomic annotations as classification features, we trained machine learning models to predict whether a human variant has an orthologue in other livestock cohorts. Variants that have cattle orthologues (with matching alleles) were used as the foreground data in the models while variants without orthologues, i.e., variants that can be lifted to the cow genome but no cattle polymorphism was found, were used as background data. Similarly, we got foreground and background datasets for pig, water buffalo, the intersection of variants found across the cattle and pig cohorts, and the cross-species cohort (cattle, pig, dog and water buffalo). For the cross-species cohorts, variants with orthologues in any of the tested livestock species were used as foreground data and variants without orthologues in all tested species were used as background data. To avoid class imbalance problems, we downsampled all background datasets to the same sizes as the foreground datasets for all models.

The feature tables were pre-processed before being used for model training. Data with missing values (found for sequence conservation scores) were discarded as they only accounted for a small proportion (1.4%) of the whole dataset. Categorical features, i.e., chromosome, consequence, allele change and 5-mer flanking sequences were encoded using different encoding methods (see Table 1). To minimize the introduction of new feature columns in the feature table and make the encoding more meaningful, a self-defined binary encoding method was used for sequence context features. We defined a dictionary for 4 bases (A: 1000, C: 0100, G: 0010, T: 0001) and mapped each base in the sequences to the corresponding binary string. The final strings for the sequences were split into binary columns and replaced the original categorical features in the feature table. We constructed three tree-based machine learning models, Random Forest[47], XGBoost[48] and CatBoost[49] using the Scikit-learn Python package[47]. Models were trained on Eddie[50], a compute cluster of the University of Edinburgh, and 2 64 GB GPUs on Eddie were used to train the CatBoost models. To enable balanced comparisons, subsets (200,000 data in total, 100,000 of which was foreground data and 100,000 background data) of the datasets for different species were used. Each subset was divided into a training set and test set at the ratio of 70% and 30%. We used 5-fold cross-validation to evaluate our models on the training sets. To improve the performance of the models, we used random search[51] and manual tuning methods for hyper-parameter tuning.

**Statistics and reproducibility**. Comparisons between two groups were conducted via Chi-Squared test, Two-sample Kolmogorov-Smirnov test and Fisher's exact test as indicated in the paper. All statistical analyses were performed using R.

**Reporting summary**. Further information on research design is available in the Nature Research Reporting Summary linked to this article.

## Data availability

The human 1000 genomes cohort genetic variants were obtained from http://ftp.1000genomes.ebi.ac.uk/vol1/ftp/release/20130502/, the pig genotypes from ftp://download.big.ac.cn/GVM/Sus_scrofa/SNP/detailed_vcf/all_SNP.vcf.gz, the dog genotypes from https://sra-pub-src-1.s3.amazonaws.com/SRZ189891/722g.990.SNP.INDEL.chrAll.vcf.1 and the cattle and water buffalo genotypes were those published in[9]. The set of annotated variants used in the machine learning analyses can be found at https://doi.org/10.6084/m9.figshare.20401851. Source data underlying Figs. 1 and 2a are provided in Supplementary Data 1 and 2, respectively. Source data underlying Fig. 3g are available on FigShare at https://doi.org/10.6084/m9.figshare.20730370. Supplementary Data 4 and 5–8 contain the source data relevant to Figs. 4–8.

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

## Acknowledgements

This work was supported by grant BB/W000288/1 and Institute Strategic Programme Grant BBS/E/D/10002070 from the Biotechnology and Biological Sciences Research Council (BBSRC).

## Author contributions

J.P. conceived the initial project idea, further developing it with R.Z., A.Ta., and M.H.; R.Z. and J.P. performed the majority of analyses with contributions from A.Ta., L.F., S.L., G.L., N.C.H., and A.Te.; L.F., S.L., G.L., and A.Te. generated the cattleGTEx data used in this study. J.P. and R.Z. wrote the initial manuscript draft with all authors contributing to the final version.

## Competing interests

The authors declare no competing interests.
