## [Peer Review File · Communications Biology]

Reviewers' comments:

Reviewer #1 (Remarks to the Author):

This is an interesting paper with a novel idea and of interest to many scientists.

On line 25 and line 426 the authors say selection maintains the polymorphisms which might lead readers to believe that the polymorphisms predate the divergence of humans from cattle and pigs. Later in the paper the authors explain that this is not what is meant but it is a pity to mislead readers early in the paper.

Later the authors argue that some polymorphisms may be maintained by selection in the current population. Even this is unlikely although possible - it is more likely that they have failed to be eliminated by selection.

I find the figures hard to read and hard to understand. In my opinion they suffer from the modern fashion to include too much detail and information in colorful figures for the sake of an impressive look. The text size is too small. Figure 1B, 2F, 4,5 and 8 I can't understand. Often the information would be clearer in a table or even a sentence or paragraph of text.

Table 1 I can't follow.

Reviewer #2 (Remarks to the Author):

The conservation of human functional variants and their effects across mammals. Zhao et al. The authors investigate genetic variants that exist across human and livestock species, and try to characterize the orthologues. This is complex and many different parameters, and therefore they use a machine learning to predict human variants that have a likely existence in livestock species. Furthermore they investigate the existence of human pathogenic variants and variants related to polygenic traits in livestock species.

It is a novel and interesting attempt to quantify what we can expect across species, and very interesting approach. I suggest the authors should also read this paper <https://doi.org/10.1371/journal.pgen.1008780>) This is (one of?) the first paper investigating human and livestock gene variants for height. Also characterizing orthologues of human and cattle variants and investigating gene length and other aspects. It continued from Bouwman et al. with a much more in depth analysis. The paper under review is a nice addition in the same promising area. Where the mentioned paper is much more a try and see approach for one trait and made the link to phenotype, in the paper under review a more general genomics based approach is used. It seems important to me to see where the results match.

It is good to see that the authors realize the importance of population size and investigate the impact in Figure 1. It is a nice and clear section to read.

Interesting to see the differences in genomics for which genes there are livestock orthologues I found Figure 2 and the related text (line 145-159) a bit cryptic. Mainly to do with shortcuts in writing, but may be also because it is complex Hence they need machine learning. However I think the clarity of the examples could be better (for a non-genomics expert, for example:

- Line 151-152 the comment "with the hypermutability...lineages" How can I see this in Figure 2A?
- Figure 2B "positive dataset" and "negative dataset" at the axis. What is meant by that?

Line 213: Where does the consequently refer to (machine learning or the section before)? It comes a bit as a surprise. Write out in full, would improve readability

For the sections under lines 212 and 271, make clear whether you investigate true orthologues in the data or predicted number of orthologues using your machine learning model.

Response to reviewers

Many thanks to the reviewers for the very helpful comments that we discuss below how we have now addressed. We believe we have managed to address all the points raised but please let us know if anything requires further alteration/clarification. As well as the changes to address the reviewers' points we noticed SNP calls for five cattle genomes had been included in the analysis beyond the 477 noted in the methods. We have therefore rerun the analyses with these five samples now excluded. As may be expected, the impact on numbers is extremely limited with generally no observable impact on figures or results. Apologies for the confusion here.

Reviewer #1 (Remarks to the Author):

This is an interesting paper with a novel idea and of interest to many scientists.

On line 25 and line 426 the authors say selection maintains the polymorphisms which might lead readers to believe that the polymorphisms predate the divergence of humans from cattle and pigs. Later in the paper the authors explain that this is not what is meant but it is a pity to mislead readers early in the paper.

Many thanks to the reviewer for highlighting these. Apologies, as the reviewer states, we don't believe that the variants predate the divergence of the species. We have reworded the text at line 25 (now line 24) to make sure this is clear:

"Models of variants linked to particular phenotypes, including metabolomic disorders and height, are preferentially shared across species"

And at line 426 (now line 445) to:

"...potentially reflects selection to preferentially maintain such variants arising in each species."

Later the authors argue that some polymorphisms may be maintained by selection in the current population. Even this is unlikely although possible - it is more likely that they have failed to be eliminated by selection.

We agree that the majority of variants that are shared across species are likely just the result of neutral processes, and this is supported by the modelling results that illustrate that they preferentially fall in regions of low purifying selection and high mutation rate. We do though believe some variants may have been maintained by selection in one or more species. For example, variants linked to stature have been shown to be under positive selection in human populations (e.g. see Turchin, M. et al. Nat Genet 44, 1015–1019 (2012)) and this phenotype has also been the target of breeding in cattle (e.g. Bouwman, A.C. et al. Nat Genet 50, 362–367 (2018)). Positive selection on variants linked to a trait in even only one species would be sufficient to lead to a preferential overlap of orthologues across species. This therefore potentially explains the disproportionate overlap of variants linked to this (and other) phenotypes across species. We have tried to clarify this at line 443:

"Consequently, although the sharing of the overwhelming majority of variants across species is likely the result of neutral processes, the disproportionate sharing of variants linked to certain phenotypes

potentially reflects selection to preferentially maintain such variants arising in each species”.

I find the figures hard to read and hard to understand. In my opinion they suffer from the modern fashion to include too much detail and information in colorful figures for the sake of an impressive look. The text size is too small. Figure 1B, 2F, 4,5 and 8 I cant understand. Often the information would be clearer in a table or even a sentence or paragraph of text.

We have gone through the figures, trying to improve their clarity, increasing font sizes and expanding their legends to provide more details. Please let us know if any figures remain unclear.

Table 1 I cant follow.

We have now expanded the table legend to try and make this table clearer.

Reviewer #2 (Remarks to the Author):

The conservation of human functional variants and their effects across mammals. Zhao et al. The authors investigate genetic variants that exist across human and livestock species, and try to characterize the orthologues. This is complex and many different parameters, and therefore they use a machine learning to predict human variants that have a likely existence in livestock species. Furthermore they investigate ,the existence of human pathogenic variants and variants related to polygenic traits in livestock species.

It is a novel and interesting attempt to quantify what we can expect across species, and very interesting approach. I suggest the authors should also read this paper <https://doi.org/10.1371/journal.pgen.1008780>) This is (one of?) the first paper investigating human and livestock gene variants for height. Also characterizing orthologues of human and cattle variants and investigating gene length and other aspects. It continued from Bouwman et al. with a much more in depth analysis. The paper under review is a nice addition in the same promising area. Where the mentioned paper is much more a try and see approach for one trait and made the link to phenotype, in the paper under review a more general genomics based approach is used. It seems important to me to see where the results match.

Many thanks for highlighting this paper. It is of clear relevance to our work, which should have been cited (which we now do at line 106). We don't though think this other paper looks at direct orthologues of variants as we believe the reviewer is suggesting? Rather it focuses on genes orthologous to those linked to height in humans. They do study cattle variants around these genes linked to stature in humans, but don't investigate variants that have an orthologue across species (as in our study). So we believe it is difficult to make a direct comparison of results across these two studies.

It is good to see that the authors realize the importance of population size and investigate the impact in Figure 1. It is a nice and clear section to read.

Interesting to see the differences in genomics f r which genes there are livestock orthologues I found Figure 2 and the related text (line 145-159) a bit cryptic. Mainly to do with shortcuts in writing, but

may be also because it is complex Hence they need machine learning. However, I think the clarity of the examples could be better (for a none genomics expert, for example):

- Line 151-152 the comment “with the hypermutability...lineages” How can I see this in Figure 2A?
- Figure 2B “positive dataset” and “negative dataset” at the axis. What is meant by that?

We have now rewritten this section to try and improve the clarity. Regarding the first point the sentence previously at line 151-152 has now been expanded to the following (now lines 149-155):

“For example, human variants with a cattle orthologue are more likely to involve a C to T change than those without a corresponding cattle orthologue (Figure 2A). C to T changes in mammalian genomes are commonly caused by the known hypermutability of CpG sites, whereby CpG sites are highly susceptible to deaminate to TpG¹⁸. The elevated mutation rates of these sites consequently likely increases the chance of the same change occurring across lineages.”

Regarding Figure 2B we have now reworded the legend to make clear what we meant by positive and negative datasets (those human variants with and without a known cattle orthologue) and have ensured we are consistent with this terminology throughout the manuscript.

Line 213: Where does the consequently refer to (machine learning or the section before)? It comes a bit as a surprise. Write out in full, would improve readability

We have now reworded this as (now line 220-224):

“Although over a million human variants have a livestock orthologue (Figure 1), the modelling results above highlight that these disproportionately fall in less conserved, and consequently most often non-functional, genomic regions. This raises the question as to how many naturally occurring livestock models of functional human pathogenic variants exist”

For the sections under lines 212 and 271, make clear whether you investigate true orthologues in the data or predicted number of orthologues using your machine learning model.

These sections were all based on true orthologues. We now clarify this at lines 231 and 283:

“Using the data from the same cow and pig cohorts we identified how often these variants overlapped an orthologous variant segregating in one of these other species”

“In total 58 of these variants had a direct orthologue segregating in either pigs or cattle”

REVIEWERS' COMMENTS:

Reviewer #1 (Remarks to the Author):

I think the authors have dealt adequately with my previous comments. However, I still find the figures hard to read.

Reviewer #2 (Remarks to the Author):

I am good with the modifications made in the document.